# An Unusual Resurgence of Human Metapneumovirus in Western Australia Following the Reduction of Non-Pharmaceutical Interventions to Prevent SARS-CoV-2 Transmission

**DOI:** 10.3390/v14102135

**Published:** 2022-09-28

**Authors:** David Anthony Foley, Chisha T. Sikazwe, Cara A. Minney-Smith, Timo Ernst, Hannah C. Moore, Mark P. Nicol, David W. Smith, Avram Levy, Christopher C. Blyth

**Affiliations:** 1Department of Microbiology, PathWest Laboratory Medicine, Perth 6009, Australia; 2Wesfarmers Centre of Vaccines and Infectious Diseases, Telethon Kids Institute, University of Western Australia, Perth 6009, Australia; 3School of Medicine, University of Western Australia, Perth 6009, Australia; 4Infection and Immunity, School of Biomedical Sciences, University of Western Australia, Perth 6009, Australia; 5Faculty of Health Sciences, School of Population Health, Curtin University, Perth 6102, Australia; 6Department of Infectious Diseases, Perth Children’s Hospital, Perth 6009, Australia

**Keywords:** non-pharmaceutical interventions, human metapneumovirus, seasonality, transmission

## Abstract

Non-pharmaceutical interventions (NPIs) to reduce SARS-CoV-2 transmission disrupted respiratory virus seasonality. We examined the unusual return of human metapneumovirus (hMPV) in Western Australia following a period of absence in 2020. We analysed hMPV laboratory testing data from 1 January 2017 to 31 December 2021. Whole-genome sequencing of selected hMPV-positive samples was performed using a tiled-amplicon approach. Following an absence in spring 2020, an unusual hMPV surge was observed during the wet summer season in the tropical Northern region in late 2020. Following a six-month delay, an intense winter season occurred in the subtropical/temperate Southern and Metropolitan regions. Compared to 2017–2019, hMPV incidence in 2021 increased by 3-fold, with a greater than 4-fold increase in children aged 1–4 years. There was a collapse in hMPV diversity in 2020, with the emergence of a single subtype. NPIs contributed to an absent 2020 season and a clonal hMPV resurgence. The summer surge and delayed winter season suggest that prevailing temperature and humidity are keys determinant of hMPV transmission. The increased incidence in 2021 was linked to an expanded cohort of hMPV-naïve 1–4-year-old children and waning population immunity. Further intense and unusual respiratory virus seasons are expected as COVID-19 associated NPIs are removed.

## 1. Introduction

Human metapneumovirus (hMPV) is a single-stranded RNA respiratory virus closely related to respiratory syncytial virus (RSV) [1]. It can cause acute respiratory tract infections in all ages, with the highest morbidity in younger children, older adults, and patients with comorbid conditions [2]. Over 14 million hMPV acute lower respiratory tract infections occur worldwide each year in children under five, with deaths infrequent [2]. In older adults, hMPV related morbidity and mortality is dependent on pre-existing co-morbidities but are comparable to influenza and RSV [3]. Mortality rates are higher in individuals that are immunocompromised, reaching over 40% following post-hematopoietic cell transplantation [4].

Immunity post-infection is incomplete, with re-infection occurring throughout life [5]. hMPV seasonality is less predictable than other respiratory viruses [6,7]. Late winter peaks are typically observed in temperate climates, frequently occurring after the RSV season [6]. Seasonality is less defined in tropical and subtropical regions, where activity is more common during spring [7].

hMPV can be divided into four major genetic lineages (A1, A2, B1 and B2) [8]. The A2 lineage is the most genetically diverse and is further divided into A2a, A2b1 and A2b2. Typically, co-circulation of different hMPV lineages occurs within the same season, with the replacement of the predominant lineage after one to two seasons [9]. The timing of infection peaks and the lineage switching patterns indicate that seasonal epidemics may be secondary to multiple introductions, with herd immunity driving viral replacement [10]. 

Globally, SARS-CoV-2 (severe acute respiratory syndrome coronavirus 2) associated non-pharmaceutical interventions (NPIs) have altered respiratory viral epidemiology [11,12,13,14]. In Western Australia (WA), border entry by international and interstate travellers was severely restricted, and two-week quarantine periods were imposed on arrivals, contributing to the successful elimination of SARS-CoV-2 between May 2020 and February 2022 [15]. Physical distancing and hand hygiene were encouraged, but large social gatherings continued, and schools remained open, apart from a short period of closure in March-April 2020. Face masks were not required outside of short sporadic periods of limited community transmission [14]. The closed borders in WA created an isolated but internally unrestricted system to observe the transmission of respiratory viruses. Influenza virus remained undetected throughout the period of restricted entry [15]. Following an absence in 2020, an inter-seasonal surge in RSV was observed in WA in late 2020 [14] and again in late 2021. In contrast, hMPV cases persisted at low levels in 2020 before increasing in early 2021. A mid-winter hMPV surge followed, with case numbers above those observed in previous seasons. 

We examined the epidemiology and genetic diversity of hMPV in WA to understand the factors underpinning the unseasonal increased detection observed in 2021. 

## 2. Materials and Methods

### 2.1. Study Location 

WA is a large, sparsely-populated state, measuring 2.6 million km^2^ with a population of 2.7 million [16]. WA spans several climatic zones, from tropical zones in the Northern regions (Kimberley and Pilbara regions) to subtropical and temperate zones in the Southern regions including the Perth metropolitan region [7]. Over 90,000 people reside in the Northern region, almost 550,000 in the Southern region and more than 2 million people (80%) in the Metropolitan region [16]. The typical winter season in WA is between June and August (weeks 22–35).

### 2.2. Laboratory Data

PathWest Laboratory Medicine WA includes the state microbiology diagnostic and reference laboratory, comprising nine metropolitan and 18 regional laboratories, performing over 16.5 million laboratory tests per year [17]. hMPV testing and detections were obtained from PathWest records (1 January 2017 to 31 December 2021). Sample collection and testing was undertaken for diagnostic purposes as per clinician request. Testing was performed using an in-house reverse transcriptase-polymerase chain reaction assay targeting the nucleoprotein gene [18].

For individuals with multiple negative hMPV tests, negative results per individual were counted once per week. Repeat positive results per individual were counted once per 3-month period. Postcode at the time of specimen collection was used to identify the individual’s place of residence. Postcode data were aggregated for analysis into three regions: Metropolitan Perth was defined as postcodes 6000–6199; Northern region, 6714–6770; and Southern regions, 6200–6710. 

The average epidemic curve was calculated using historical PathWest data between 2013–2019 as previously described [14]. The Moving Epidemic Method (MEM) was used to establish season onset and offset thresholds using data from the same period [19]. MEM uses detections per week to calculate pre- and post-epidemic thresholds, and has been applied to several respiratory viruses, including influenza and RSV [19]. MEM also establishes season intensity thresholds, set at medium, high and very high levels. Epidemic thresholds and intensity levels were rounded up to the nearest whole integer. 

MicroReact, an online application, was used to generate temporospatial visualizations of hMPV detections (Wellcome Sanger Institute, Hixton, UK) [20]. The timing was determined by the date of the specimen collection, and location was based on the individual’s postcode converted to longitude and latitude [21]. 

Ethical approval was granted by the Western Australian Child and Adolescent Health Service (GEKO: 42341). 

### 2.3. Whole Genome Sequencing and Statistical Analysis

A purposively sampled selection of hMPV-positive routine diagnostic specimens was chosen ensuring geographical and temporal representation from 2017 to 2021. Nucleic acid was extracted, and tiled amplicon sequencing was performed using established protocols [22,23]. In brief, four primer sets were used to amplify overlapping amplicons spanning the hMPV genome. The products were sequenced on an Illumina iSeq 100 platform (Illumina, San Diego, CA, USA). The generated sequences were analysed with reference sequences from the hMPV global dataset using the processes previously outlined [23]. An additional 200 coding complete genome sequences representing the current global diversity of hMPV were obtained from GenBank [24].

A dated phylogeny was reconstructed using BEAST 1.10.4 and the Bayesian Markov chain Monte Carlo (MCMC) method [25]. An alignment of 119 sequences, representing a subset of the global hMPV dataset was generated using MAFFT 7.450 [26]. The general time reversible (GTR)+Ι+Γ_4_ substitution model using jModelTest 2.1.10 [27]. A Bayesian skyline model with a relaxed uncorrelated log normal molecular clock was used [28,29]. The MCMC was run for 200 million generations, sampling every 20,000 steps and appropriate levels of mixing were ensured using Tracer 1.7.1 [30]. The maximum clade credibility (MCC) tree was determined using TreeAnnotator 1.10.4 and visualised using FigTree 1.4.2 [25].

Testing and positive detections for hMPV were compared by calendar year. Temporal figures were constructed using ISO week. Statistical analyses were performed using R [31]. Version 2.16 of the MEM R package was utilized [32]. Population data from 2020, obtained from the Australian Bureau of Statistics, were used as denominators to calculate hMPV detection rates [16], reported incidence rates per 100,000 population, and incidence rate ratios (IRR). 

## 3. Results

Between 1 January 2017 and 2 January 2022, 119,651 hMPV tests were performed and 3964 (3.3%) were positive. The season threshold was determined using MEM and set as ≥20 detections per week for onset and <21 for offset. The average hMPV season commenced at week 35 (95% confidence interval [CI], 34–39 weeks), lasted 13 weeks (95% CI, 12–16 weeks) and captured 68% of the year’s total detections (Appendix A). hMPV season onset occurred consistently towards the end of winter, with the peak observed in spring. The 2017 season was the shortest observed, while the 2019 season started earlier and lasted longer than the preceding years hMPV was detected in WA in early 2020. There was a decrease in detections following implementation of SARS-CoV-2-associated NPIs (Figure 1). Very low hMPV detections were observed throughout 2020, with an absent winter season. An out-of-season increase was observed from week 48, December 2020 (southern hemisphere summer), clustered in the Northern Region of WA (Figure 1, Appendix A). Detection numbers crossed the MEM season onset threshold on week one of 2021 and lasted until week six. For analytical purposes, this was labelled as the 2021 minor season. A subsequent drop in detections was temporally associated with a short period of increased NPIs. However, these NPIs, including a requirement to wear facemasks, were instigated only in the Metropolitan and Southern regions. Population movement restrictions were in place in the Northern region, but facemasks were not mandated [33]. Low detections numbers continued in the Northern region of WA, with a smaller decrease from week 16 associated with an additional and extended four-week period of increased NPIs in the Metropolitan and Southern regions.

hMPV was consistently detected in the Metropolitan region from week 21 of 2021 (Figure 1 and Appendix A). A sharp increase in detections was observed on week 26, contemporaneous with a short period of increased NPIs in this region. hMPV was commonly detected in the Southern region from week 30 of 2021 (Figure 1 and Appendix A). State-level detections peaked at week 33 dropping to low levels by week 41, 2021. 

This “primary” 2021 hMPV season commenced on week 26, earlier than previous years. This season lasted 16 weeks, accounting for 78% of the year’s total detections (Table 1). There were 1474 detections during this season, almost three times more than the next highest year (2019). There were ten weeks in the primary season above the MEM very high threshold (≥64 detections per week), a threshold not exceeded in other years (2013–2019). Detections were initially predominantly in children 0–4 years of age, shifting to those ≥16 years of age as the season progressed (Appendix A). By comparison, age groups were more evenly dispersed throughout the 2017–2019 seasons. 

There were 1879 hMPV detections in 2021, more than in any other year. Comparing 2021 and pre-SARS-CoV-2 years (2017–2019), the incidence of hMPV in WA increased 3-fold from a baseline average of 23.8 per 100,000/year (95% CI, 22.8–24.9) to 70.6 (95% CI, 67.5–73.8 (IRR 3 [95% CI, 2.8–3.2] (Table 2 and Appendix A). 

During the baseline period between 2017 and 2019, the high proportion of detections were from the more populous Metropolitan region (>60%). However, the highest hMPV incidence was in the Northern region (Table 2) with an incidence rate of 66.4/100,000. hMPV incidence dropped in the Metropolitan and Southern Region in 2020. In contrast to other regions, hMPV incidence in the Northern region increased, linked to the increase in detections towards the end of the year (incidence rate ratio, 1.2 [95% CI, 0.9–1.6]) (Figure 1, Appendix A). 

The incidence of hMPV increased in all regions in 2021 (Table 2 and Appendix A). The most significant change was observed in the Northern region, increasing almost 6-fold (IRR 5.9, 95% CI, 4.9–7.0) compared to the mean incidence between 2017 and 2019. The Southern region also experienced a more than three-fold increase in incidence (IRR 3.6, 95% CI, 3.2–4.1).

Before 2020, more than 50% of hMPV detections were in individuals ≥16 years of age, with one in four detections in older adults (≥65 years). The mean hMPV incidence between 2017–2019 was highest in infants under 12 months of age (274.1/100,000, 95% CI 242.2-308.7), followed by those aged 1–4 years (91.7/100,000, 95% CI 82.7–101.4) and those 65 years and older (43.6/100,000 [95% CI, 40.0–47.5]; Table 2). In 2021, the incidence of hMPV increased in all age groups. The most significant increases were seen in children older than 12 months; a four-fold increase in the 1–4 year age group (IRR 4.2, 95% CI 3.7–4.8) and more than three-fold in the 5–15 year (IRR 3.8, 95% CI, 3.0–4.7).

Testing for hMPV increased over time, peaking in 2020 and 2021 (Table 1). Increases in testing were associated with the emergence of SARS-CoV-2 in 2020 and periods of increased NPIs in 2021 (Figure 1). Increases in hMPV testing were noted across the regions (Appendix A). The largest increase in testing was temporally associated with the first detection of SARS-CoV-2 in WA on 21 February 2020. 

However, overall hMPV test percentage positivity was higher in 2021 than in previous years (*p* < 0.001; Table 1 and Figure 2). The increase in hMPV testing post SARS-CoV-2 emergence was not uniform. Significant testing increases were observed in those 1–4 years of age with smaller increases observed in those aged <12 months, 5–15 years and ≥65 years (Figure 2). The 2021 hMPV testing percentage positivity for older adults was similar in 2017–2019 (45–64 years old and ≥65 years). There was a significant increase in hMPV 2021 percentage positivity in those aged <12 months, 1–4 years and 5–15 years compared to (2017 to 2019, *p* < 0.001), with the proportion of positive tests exceeding 10% in the 1–4-year-old group (Figure 2).

### Molecular Epidemiology of hMPV

A total of 53 coding complete hMPV genomes were generated. Genomes were distributed into either hMPV lineage A (n = 32) or B (n = 21). All A lineage viruses belonged to the A2.b sub-lineage (Figure 3). Of the 21 B lineage viruses detected, 11 were positioned in the B.1 sub-lineage and 10 in the B.2 sub-lineage (Appendix A). None of the hMPV genomes belonged to either A.1 or A.2a sub-lineages. 

Genomes generated from samples taken before the emergence of SARS-CoV-2 were distributed throughout the phylogeny, belonging to either B.1, B.2 or A.2b lineage, with multiple lineages observed during a single season (Figure 3 and Appendix A). In contrast, following the reduction in SARS-CoV-2-associated NPIs, a novel monophyletic clade of hMPV A.2.2 viruses emerged, here designated as A2.bWA. hMPV A2.bWA was first detected in July 2020 in the northern region of WA and subsequently became the single strain detected during the 2021 minor season in the Northern region and the subsequent primary season in the Metropolitan and Southern regions. A2.bWA viruses were most closely related to a 2019 USA hMPV A.2.2 sequence (GenBank accession number MT118691) and 2019 hMPV A2.b viruses sampled in WA which was sequenced as part of this study. 

From the molecular clock analyses, it is estimated that the A2.bWA viruses emerged in May 2019 (95% highest probability density [HPD], September 2018 to November 2019 (Figure 3B). The evolutionary rate of the A2.bWA clade was estimated to be approximately 1.27 × 10^−3^ substitutions/site/year (95% HPD, 1.77 × 10^−3^–8.55 × 10^−4^ substitutions/site/year). Sequence analysis identified numerous non-synonymous changes in the fusion (12), small hydrophobic (4) and attachment glycoprotein genes (13) of unclear significance.

## 4. Discussion

We assessed the temporal and seasonal patterns of hMPV testing and detections across the WA population over five years. Prior to COVID-19, seasonal epidemics were observed, with onset in late winter peaking in spring. SARS-CoV-2-associated NPIs were associated with a reduction in hMPV detections, an absent spring season in 2020 and an atypical summer surge of hMPV during the Northern region’s summer wet season. Following a delay of six months, hMPV incidence increased in the Metropolitan region in the winter of 2021, first detected in children ≤4yrs and subsequently in older populations. This season had more detections than any previously observed. Overall, hMPV incidence almost tripled in 2021 compared with baseline period. The most significant increase was observed in those aged 1–4 years. Prior to 2020, phylogenetic analyses demonstrated multi-lineages circulating in a single season. In 2021, a collapse in hMPV diversity was observed, with only a single subtype detected state-wide.

This study confirms that hMPV, similar to influenza and RSV [15], is susceptible to NPIs. Before 2020, hMPV was detected in WA throughout the year, with season onset towards late winter, peaking in spring. These seasonal patterns in WA were significantly disrupted following the instigation of SARS-CoV-2-associated NPIs. These measures introduced in early 2020 contributed to the reduction in endemic transmission and the elimination of multiple hMPV lineages.

Despite a return to minimal internal restrictions by mid-2020 and persistent low-level hMPV detections, the expected seasonal peak remained absent. The state border remained closed during this period, with a quarantine requirement on entry. This closed border limited the opportunity for introduction of new hMPV lineages into the local population. Phylogenetic analysis of prior seasons infers that viral migration is an important driver of WA hMPV seasonality. Before 2020, multiple hMPV lineages co-circulated each year, some persisting into subsequent seasons while others were replaced by new lineages. We conclude that maintaining closed state borders prevented the introduction of variants from the global pool of hMPV viruses, contributing to a collapse in viral diversity and an absent spring season. This mirrors the pattern of RSV activity that was seen across the same time period [23].

hMPV detections subsequently increased in the Northern tropical region in late 2020 and early 2021. This timing coincided with the area’s wet season and a period of above-average rainfall in this region [30]. Periods of rainfall and high humidity in tropical regions are associated with higher rates of hMPV transmission [34]. However, a summer season has not previously been observed in this region. It is plausible that introduction of hMPV following a prolonged absence during favourable environmental conditions contributed to this outcome.

Due to its size, WA spans tropical to temperate climate regions. Most people reside in the Metropolitan region in Southern WA, which experiences a sub-tropical to temperate climate [7]. Despite a similar time from the last hMPV season as the Northern region and higher population density in these regions favouring transmission, hMPV detections were infrequent in the Metropolitan and Southern regions until winter 2021. The less favourable environmental conditions may have prevented sustained hMPV transmission in these areas, supplemented by short periods of increased NPIs in these regions. The primary 2021 season began in winter, as the environmental conditions became cooler and more humid, favouring hMPV transmission [34]. The subsequent season was more intense than any previously observed, representing almost 80% of the year’s total detections.

Overall, hMPV incidence in 2021 was almost threefold higher than in 2017–2019. Although incidence rose across all age groups, the increase in rates in children ≤15 years was a key contributor. The most significant increase in detection rates and absolute numbers occurred in the 1–4 year age group. The absent season and low inter-seasonal levels in 2020 produced an 18-month period without hMPV circulation. We propose that this contributed to an expanded hMPV naïve cohort in this group and subsequent first infection at an older age. This pattern of increased infection in the 1–4 year age group was also observed during a delayed inter-seasonal surge in RSV in WA following an absent winter season [14].

hMPV incidence also increased in older individuals, possibly linked to the 18 months of low circulating levels. As immunity post-infection is incomplete [5], the reduction in circulating hMPV may have contributed to the waning of pre-existing immunity in older age groups. It is possible that the emergence of a new hMPV sub-lineage with altered transmissibility and pathogenicity may have contributed to this increase. However, sequence analysis of the 2021 strain did not identify any known markers. Further, although not detected in WA until June 2020, genomic analysis suggests the emergence of the 2021 strain in February 2019. Given the closed borders in 2020, it is credible that this hMPV strain was circulating in the WA during the 2019 season but remained undetected due to incomplete genomic sampling. However, sufficient sampling was performed to identify a dominant subtype, denoting that specific viral determinant of transmission or pathogenicity is unlikely to be a central driver of the hMPV resurgence. In contrast, pre-existing exposure would generate strain-specific immunity, blunting the increase in infections in older age groups.

Expanded testing was an important factor driving some of the increase in 2021 hMPV detection. Unsurprisingly, testing practices shifted significantly in response to the emergence of SARS-CoV-2, with notable additional increases during periods of increased NPIs. Despite these changes, test percentage positivity was high in 2021. In older adults, it remained similar to previous years. Further, percentage positivity rose substantially in children ≤15 years, reaching over 10% in the 1–4 year group. Although expanded testing may have identified infections that would have been under-ascertained, the persistently raised percentage positivity reinforce that the increased detections in 2021 were based on a genuine rise in cases.

As the global focus has shifted to living with SARS-CoV-2, the return to pre-pandemic behaviours, including increased population mobility, will facilitate global respiratory virus transmission. With the opening of borders, there are concerns that the introduction of multiple strains of hMPV and other respiratory viruses into susceptible populations will result in a multimodal respiratory virus epidemic placing a considerable strain on the healthcare system. Other regions will face similar challenges, depending on previous and contemporaneous NPIs.

This analysis has several potential limitations. hMPV testing was performed as part of diagnostic purposes and represent biased sampling of the general population. Baseline differences in health-seeking behaviours between ages and regions may have contributed to differences in detection rates, especially in Northern WA [35]. This study has not directly assessed changes in health-seeking behaviours and clinician testing thresholds following the emergence of SARS-CoV-2. hMPV testing and detection data were obtained from a single pathology provider. Although this provider has a broad state-wide footprint, this limitation may introduce biases into the data collected, including under ascertainment of hMPV in the metropolitan region where other pathology providers are active. Furthermore, decreased circulation of other respiratory viruses may have inflated hMPV testing percentage positivity. Finally, despite selection of a representative samples, the number of hMPV sequences was moderate compared to the number of detections.

## 5. Conclusions

NPIs that successfully curbed transmission of SARS-CoV-2 contributed to the collapse of hMPV diversity in WA and an absent 2020 season. The subsequent resurgences of hMPV have highlighted that, similar to influenza and RSV, population movement, host susceptibility, and climate conditions are key determinants of hMPV transmission. The increased incidence in 2021 is linked to an expanded cohort of hMPV naïve 1–4-year-old children and waning population immunity. Further intense and unusual respiratory virus seasons are expected as international travel resumes and local NPIs are further reduced. Further consideration needs to be given to the significance of the restricted genomic diversity of the re-emerging hMPV and RSV epidemics, and whether this observation be restricted to the pneumoviruses or replicated in other respiratory viruses.

## Figures and Tables

**Figure 1 viruses-14-02135-f001:**
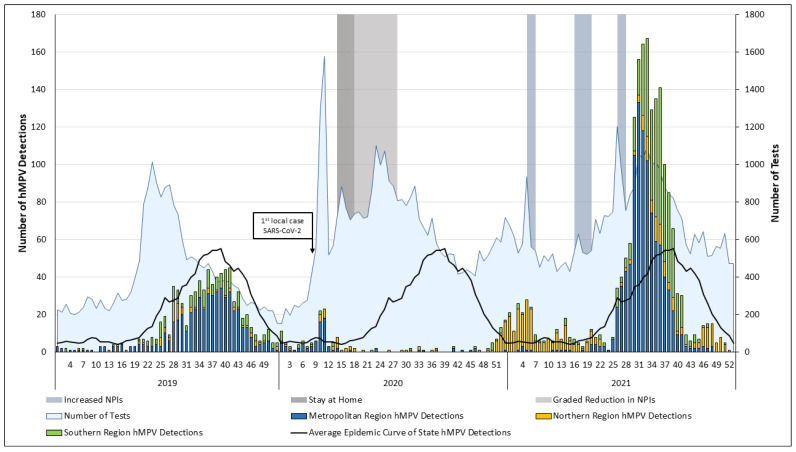
hMPV tests and detections per week by region between week one, 2019 and week 52, 2021. The average epidemic curve is based on hMPV detections in PathWest Laboratory Medicine between 2013 and 2019, inclusive. The first local case of SARS-CoV-2 was in week eight of 2020. State-wide stay at home order (weeks 14–17 of 2020) was followed by sequential lifting of restrictions (18–26 weeks) with gatherings of 10 allowed at week 18, 20 at 21 weeks and 100 at week 24. Periods of increased NPIs included mask requirements in the Metropolitan and Southern region (weeks 5 and 6, weeks 16 to 19 and 26 to 27) in 2021. hMPV, human metapneumovirus; NPI, non-pharmaceutical intervention; SARS-CoV-2, severe acute respiratory syndrome coronavirus 2).

**Figure 2 viruses-14-02135-f002:**
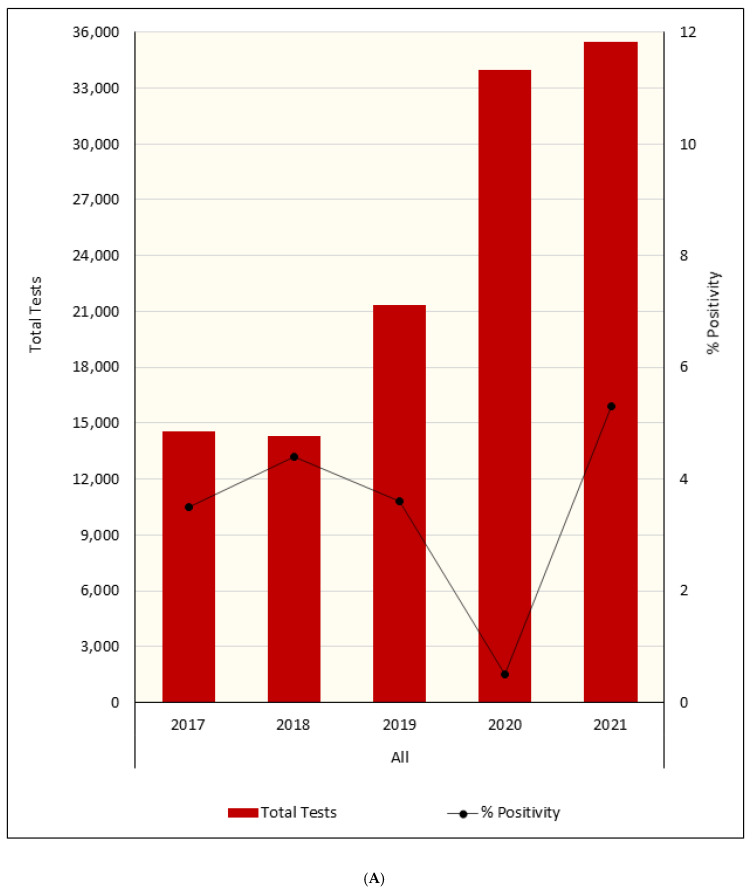
(**A**) Total hMPV tests and percentage positivity per year in Western Australia; (**B**) Total hMPV tests and percentage positivity per year by age groupings; (**C**) Total hMPV tests and percentage positivity per year by age groupings by region.

**Figure 3 viruses-14-02135-f003:**
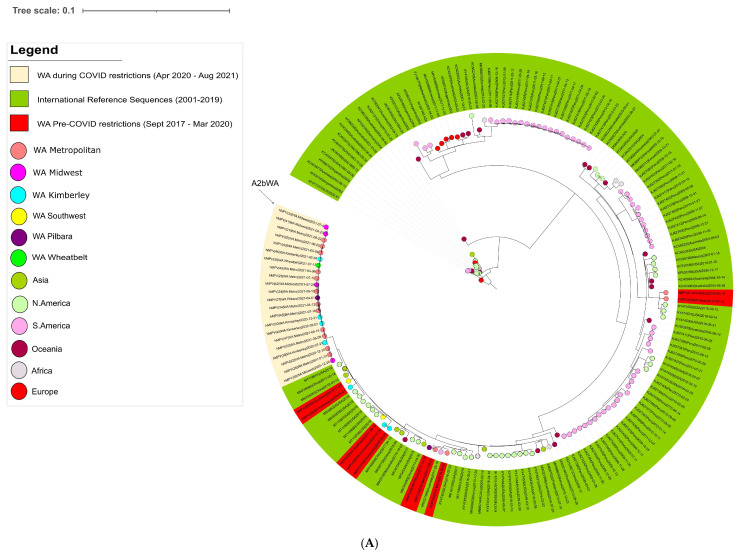
(**A**) Phylogenetic analysis of hMPV lineage A viruses sampled in Western Australia (WA) between 2017 and 2021. WA hMPV genomes sampled during the period after the instigation of SARS-CoV-2 related non-pharmaceutical interventions are shaded pale orange. WA hMPV genomes sampled before this period are shaded red. Reference sequences download from GenBank are shaded green. The scale represents the number of nucleotide substitutions per site; (**B**) Maximum clade credibility phylogeny of 153 complete genomes of hMPV, with estimated date and 95% posterior densities given for the node of interest (shaded purple) described in the text.

**Table 1 viruses-14-02135-t001:** Comparison of hMPV testing and detections in Western Australia by calendar year between 2017 to 2021. Season duration derived from Moving Epidemic Method onset and offset thresholds. For 2021, only the primary season was included in the season figures. Metropolitan region based on postcode 6000–6199. 95% CI, 95% Confidence Interval; hMPV, human metapneumovirus; N/A, not applicable.

	Year				
	2017	2018	2019	2020	2021
hMPV testing					
Detections; n	508	626	769	182	1879
Total tests; n	14,553	14,293	21,339	33,959	35,507
% Positive (95% CI)	3.5 (3.2–3.8)	4.4 (4.1–4.7)	3.6 (3.4–3.9)	0.5 (0.5–0.6)	5.3 (5–5.5)
% Metropolitan (95% CI)	69 (64–73)	62 (58–66)	67 (64–71)	44. (37–52)	51 (49–53)
Age band detections	n (% year total)				
<12 months	79 (16)	95 (15)	97 (13)	18 (10)	226 (13)
1–4 years	107 (21)	142 (23)	132 (17)	23 (13)	533 (31)
5–15 years	27 (5)	51 (8)	67 (9)	14 (8)	182 (10)
16–64 years	155 (30)	189 (30)	229 (30)	93 (51)	601 (31)
≥65 years	140 (28)	149 (24)	244 (32)	34 (19)	337 (20)
Season					
Duration (weeks)	7	12	16	N/A	16
Timing (week range)	35 to 41	31 to 42	28 to 43	N/A	26 to 41
Detections; n (% of year total)	178 (35%)	361 (58%)	551 (72%)	N/A	1474 (78%)

**Table 2 viruses-14-02135-t002:** Comparison of hMPV mean incidence per 100,000 population group (with 95% CI) between 2017 to 2019 with 2020 and 2021, expressed as an incidence rate ratio. 95% CI, 95% confidence interval; IRR, incidence rate ratio.

	Year		
	2017–2019	2020	2021
	Incidence (95% CI)	Incidence rate ratio (95% CI)	Incidence rate ratio (95% CI)
All	23.8 (22.8 to 24.9)	0.3 (0.3 to 0.3)	3 (2.8 to 3.2)
By region	Incidence (95% CI)	Incidence rate ratio (95% CI)	Incidence rate ratio (95% CI)
Metropolitan region	20.6 (19.5 to 21.8)	0.2 IRR (0.2 to 0.2)	2.3 IRR (2.1 to 2.5)
Northern region	66.4 (57.1 to 76.8)	1.2 IRR (0.9 to 1.6)	5.9 IRR (4.9 to 7.0)
Southern region	28.7 (26.2 to 31.4)	0.2 IRR (0.1 to 0.3)	3.6 IRR (3.2 to 4.1)
By age group	Incidence (95% CI)	Incidence rate ratio (95% CI)	Incidence rate ratio (95% CI)
<12 months	274.1 (242.2 to 308.7)	0.2 IRR (0.1 to 0.3)	2.5 IRR (2.1 to 3)
1–4 years	91.7 (82.7 to 101.4)	0.2 IRR (0.1 to 0.3)	4.2 IRR (3.7 to 4.8)
5–15 years	12.8 (10.8 to 15.1)	0.3 IRR (0.15 to 0.5)	3.8 IRR (3.0 to 4.72)
16–64 years	11.1 (10.3 to 12.2)	0.5 IRR (0.4 to 0.6)	3.2 IRR (2.8 to 3.5)
≥65 years	43.6 (40 to 47.5)	0.2 IRR (0.1 to 0.3)	1.9 IRR (1.7 to 2.2)

## Data Availability

Restrictions apply to the availability of these data. Data are available through Western Australia Health subject to appropriate ethics and governance approval.

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
