# Peer review of "An Unusual Resurgence of Human Metapneumovirus in Western Australia Following the Reduction of Non-Pharmaceutical Interventions to Prevent SARS-CoV-2 Transmission"

_viruses, 2022, doi:10.3390/v14102135_

Round 1
Reviewer 1 Report
Foley et al. describe a 5-yrs long epidemiological survey of hMPV performed in a wide Australian geographic area before and after SARS-CoV-2 pandemic.
The research group is expert in the field, having already published similar evaluations involving other viruses or environmental settings. The research is well conducted, the dataset is quite large, thus allowing to have clear-cut results.
Results are interesting and add a piece in the intricate puzzle of respiratory virus circulation patterns. Of note, the emerging of a single new viral sublineage after COVID-19 pandemic.
Comments:
- MM section: samples analyzed were from hospitalized people, symptomatic individuals or are random sampling for a population survey? From the text it looks like samples were collected for diagnostic purposes. This is an important point since increase in detection rate and absolute number may be due to augumented hMPV virulence or loss of immunity in the population and my not represent a real increase in viral circulation among non-symptomatic persons. Please specify.
- Figure 2: legend has to be a bit more informative. Please correct.
- Discussion, line 354: XYZ? Please correct the error
- Discussion is mainly focused on circulation pattern and climate regions. I would rather dscuss more extensively the emergence of a new sublineage and I would speculate about strain-specific viral pathogenicity
Among limitation I suggest to add lack of information about asymptomatic general population
Author Response
Dear Reviewer
Thank you for reviewing our submission. We appreciate the positive feedback. In response to the recommendations:
- Sampling: We agree with the importance of this point and have made changes to the document to reflect that samples were collected, and tested for diagnostic purposes as per physician request (line 88 in tracked changes version) and included in our limitations that this type of sampling will bias the dataset
- Figure 2 legend: altered to improve clarity. The authors are happy to make further alterations if necessary, guided by the reviewer’s recommendations.
- Discussion line 354: Thank you for picking up this error
- Add re-emergence of new sub lineage, strain specific viral pathogenicity: we have added this component to the discussion (line 416-425 of tracked changes version)
With thanks,
Dr David Foley on behalf of the author group
Reviewer 2 Report
--
Review comments for manuscript ID: " viruses-1915796 ", entitled "An unusual resurgence of human metapneumovirus in Western Australia following the reduction of non-pharmaceutical interventions to prevent SARS-CoV-2 transmission" of the journal "Viruses".
--
General Comments:
This is a nice and well-conducted epidemiological study which examined an unusual reemergence of human metapneumovirus in Western Australia following the removal of SARS-CoV-2 NPIs. I consider their method reasonable and sophisticated, and the results are presented logically and consistently. The analyzing outcomes support the main results with biologically and clinically appropriate settings. Thus, I would recommend publishing it after the following concerns are considered.
--"Following an absence in 2020, an inter-seasonal surge in RSV was observed in WA in late 2020 [12] and again in late 2021". I wonder whether the authors meant COVID-NPIs had caused an immediate impact on the hMPV transmission. The authors should provide explicit epidemiological evidence to support this assertion. Also, it is unclear why there was a sudden increase in hMPV transmission in early 2021 despite most of the NPIs being in place during that period. What part of the COVID-NPIs mainly affect the hMPV transmission in WA?
--Basic epidemiological background for the hMPV, regarding the morbidity and mortality in WA or worldwide, should be provided in the introduction section. https://www.cdc.gov/surveillance/nrevss/hmpv/clinical.html, https://doi.org/10.1038/srep27730, https://doi.org/10.1128/CMR.00014-06.
--The clarity of all the figures should be improved, especially the supplementary figures.
--Results presentations should be reframed. I appreciate the authors have done a very nice discussion, mainly focusing on the technical part. I would be more appreciative if the authors could elaborate more on the epidemiology and public health implications.
Author Response
Dear Reviewer,
Thank you for reviewing our submission. We appreciate the time you have taken to provide feedback. In response to the recommendations:
- "Following an absence in 2020, an inter-seasonal surge in RSV was observed in WA in late 2020 [12] and again in late 2021". I wonder whether the authors meant COVID-NPIs had caused an immediate impact on the hMPV transmission.
Apologies if we misinterpret the intended direction of this feedback.
This sentence refers to our previous observations of the inter-seasonal surge of RSV, essentially setting the scene. As discussed earlier in the introduction, the “heaviest” NPIs were introduced in WA in early 2020 but were quickly reduced, with life returning to normal within the state. However, the state border remained closed, with a quarantine requirement on entry. This maintenance of a closed state border created an “isolated but internally unrestricted system to observe the transmission of respiratory viruses.” We have altered the manuscript to underline this important detail (line 66-67 of the tracked changes version)
- I wonder whether the authors meant COVID-NPIs had caused an immediate impact on the hMPV transmission. The authors should provide explicit epidemiological evidence to support this assertion.
We try to provide this evidence in figure 1 (during and following the “grey” periods), noting the drop in hMPV detections. Further, our genomic analysis demonstrated a collapse in viral diversity and persistence of A2.bWA, first detected in early July (during the period of closed borders).
- Also, it is unclear why there was a sudden increase in hMPV transmission in early 2021 despite most of the NPIs being in place during that period. What part of the COVID-NPIs mainly affect the hMPV transmission in WA?
The internal NPIs (facemasks) in 2021 were implemented in the Metropolitan and Southern regions for brief periods (see figure 1), not the Northern region where the majority of detections occurred in early 2021. We postulate that the combination of suitable localised environmental conditions and susceptible hosts drove this increase in this period (lines 385-394 in discussion).
Interestingly, the later 2021 “primary season” did start in the Metropolitan region during a period of increased NPIs (facemasks for >12). As seen in supplementary figure 3, detections were predominantly in children 0-4yrs shifting to older age groups as the season progressed and the NPIs were reduced. We did not include this in the discussion due concerns regarding length of the manuscript. We would be happy to add this discussion point if of interest to the Reviewer and Editors
- The clarity of all the figures should be improved:
High quality figures have been generated in TIFFs for publication, if accepted. We are happy to make specific adjustments to figures if the Reviewer or team at Viruses have specific recommendations
- Additional basic epidemiology
We have altered the introduction to include additional morbidity and mortality data (line 36-44 of tracked changes version)
- Reframing results
Although non-pharmaceutical measures resulted in a collapse in hMPV viral diversity in early 2020, we don’t believe that NPIs themselves are a key part of this story. WA NPIs essentially created a closed system to observe the behaviour of viruses, combining a limited opportunity for “new” viruses to be introduction by closing the state border and minimal restrictions internally to impair transmission. We have tried to address the public health implications in the discussion (lines 437-443) and the conclusion (463-467). We would welcome further guidance on how the Reviewer would suggest the results be reframed, if required.
With thanks,
Dr David Foley on behalf of the author group